# A Systematic Review to Evaluate Patient-Reported Outcome Measures (PROMs) for Metastatic Prostate Cancer According to the COnsensus-Based Standard for the Selection of Health Measurement INstruments (COSMIN) Methodology

**DOI:** 10.3390/cancers14205120

**Published:** 2022-10-19

**Authors:** Maria Monica Ratti, Giorgio Gandaglia, Elena Silvia Sisca, Alexandra Derevianko, Eugenia Alleva, Katharina Beyer, Charlotte Moss, Francesco Barletta, Simone Scuderi, Muhammad Imran Omar, Steven MacLennan, Paula R. Williamson, Jihong Zong, Sara J. MacLennan, Nicolas Mottet, Philip Cornford, Olalekan Lee Aiyegbusi, Mieke Van Hemelrijck, James N’Dow, Alberto Briganti

**Affiliations:** 1Department of Medicine and Surgery, Vita Salute San Raffaele University, 20132 Milan, Italy; 2Department of Clinical and Health Psychology, IRCCS San Raffaele Hospital, 20132 Milan, Italy; 3Unit of Urology/Division of Oncology, URI, IRCCS San Raffaele Hospital, 20132 Milan, Italy; 4Translational and Oncology Research (TOUR), Faculty of Life Sciences and Medicine, King’s College London, London WC2R 2LS, UK; 5Academic Urology Unit, University of Aberdeen, Aberdeen AB24 3UE, UK; 6MRC North West Hub for Trials Methodology Research, University of Liverpool, a Member of Liverpool Health Partners, Liverpool L7 8XP, UK; 7Real World Evidence, Global Medical Affairs Oncology, Whippany, NJ 07999, USA; 8Department of Urology, University Hospital, 42055 St. Etienne, France; 9Liverpool University Hospitals NHS Trust, Liverpool L69 3GA, UK; 10Centre for Patient-Reported Outcomes Research, College of Medical and Dental Sciences, University of Birmingham, Birmingham B15 2TT, UK

**Keywords:** prostatic neoplasms, COSMIN, PROMs, quality of life, erectile dysfunction, metastatic, prostate cancer, core outcome set

## Abstract

**Simple Summary:**

Metastatic prostate cancer (mPCa) is one of the most common solid tumors in men and both the disease and the treatments affect patients’ quality of life (QoL). Patient-reported Outcome Measurements (PROMs) are important to assess the patient’s subjective experience with disease and treatment. Our aim is to appraise, compare, and summarize the psychometric properties of Patient-reported Outcome Measures (PROMs). Our findings can improve patients’ care and their quality of life during treatment and the disease path.

**Abstract:**

Introduction: Patient-reported outcome measures (PROMs) represent important endpoints in metastatic prostate cancer (mPCa). However, the clinically valid and accurate measurement of health-related quality of life depends on the psychometric properties of the PROMs considered. Objective: To appraise, compare, and summarize the properties of PROMs in mPCa. Evidence acquisition: We performed a review of PROMs used in RCTs, including patients with mPCa, using Medline in September 2021, according to the COnsensus-based Standards for the selection of health Measurement INstruments (COSMIN) criteria. This systematic review is part of PIONEER (an IMI2 European network of excellence for big data in PCa). Results: The most frequently used PROMs in RCTs of patients with mPCa were the Functional Assessment for Cancer Therapy—Prostate (FACT-P) (*n* = 18), the Brief Pain Inventory—Short Form (BPI-SF) (*n* = 8), and the European Organization for Research and Treatment of Cancer quality of life core 30 (EORTC QLQ-C30) (*n* = 6). A total of 283 abstracts were screened and 12 full-text studies were evaluated. A total of two, one, and two studies reported the psychometric proprieties of FACT-P, Brief Pain Inventory (BPI), and BPI-SF, respectively. FACT-P and BPI showed a high content validity, while BPI-SF showed a moderate content validity. FACT-P and BPI showed a high internal consistency (summarized by Cronbach’s α 0.70–0.95). Conclusions: The use of BPI and FACT-P in mPCa patients is supported by their high content validity and internal consistency. Since BPI is focused on pain assessment, we recommend FACT-P, which provides a broader assessment of QoL and wellbeing, for the clinical evaluation of mPCa patients. However, these considerations have been elaborated on in a very limited number of studies. Patient summary: In this paper, we review the psychometric properties of PROMs used with patients with mPCa to find the questionnaires that best assess patients’ QoL, in order to help professionals in their intervention and improve patients’ QoL. We recommend the use of BPI and FACT-P for their high content validity and internal consistency despite the limited number of studies considered.

## 1. Introduction

Prostate cancer (PCa) is one of the most common solid tumors with more than an estimated 1,414,000 new cases diagnosed in the year 2020 and 375,000 deaths worldwide [1]. Although the introduction of PSA and early detection strategies have decreased the rate of advanced and metastatic disease, a non-negligible proportion of men harbor metastases at diagnosis or progress after primary treatment [2]. Given the slow progression of PCa, there are a range of possible life expectancies for men with metastatic hormone-sensitive disease (mHSPC). Life expectancy has improved from a median of 3 to 7 years over the last decades. As a result, a better understanding of how both the disease and the treatments affect the quality of life (QoL) is essential to guide the improvement in patient care. In addition to oncological endpoints and clinicians’ reported outcomes such as the progression or development of a castration-resistant state, Patient-reported Outcome Measurements (PROMs) are important to assess the patient’s subjective experience with disease and treatment. Moreover, they can provide clinically relevant, patient focused, and reliable perspectives on quality of life and the patient’s symptom experience.

Various generic and disease-specific PROMs have been developed and can be employed to assess the quality of life of PCa patients. In this context, the PIONEER consortium recently reported which PROMs should be used for the evaluation of patients with localized disease according to the COnsensus-based Standards for the selection of Health Measurement INstruments (COSMIN) methodology [3]. Based on of this systematic review, the EORTC QLQ-30 and EORTC-PR25 were recommended to measure the core domains of urinary, bowel, sexual function, hormonal symptoms, and health-related quality of life (HRQoL) both in research and routine care settings. However, due to the use of a stringent methodology, these results can only be applied to patients with clinically localized disease. Under these premises, we aimed to assess which PROMS should be used to evaluate patients with mPCa in both the clinical and research settings. In particular, the quality of studies reporting on the psychometric properties of PROMs used in randomized controlled trials that focused on patients with metastatic PCa (mPCa) was evaluated with the aim to provide perspectives on the most suitable instruments that should be adopted for the assessment of patients with mPCa [4,5].

## 2. Evidence Acquisition

### 2.1. Search Strategy

A systematic review of the literature was performed to identify any PROMs developed or applied in patients with mPCa between January 2014 and January 2021. The systematic review was conducted according to a predefined protocol based on the COSMIN guidance [4,6]. Relevant studies were identified by conducting searches of Medline (PubMed). To identify a comprehensive list of PROMs used in mPCa intervention effectiveness trials, we updated the systematic review by MacLennan and colleagues [7] and used the same inclusion/exclusion criteria.

The review is reported in accordance with the PRISMA statement [8]. For this systematic review, instruments/measures to record HRQoL are referred to as PROMs, which are questionnaires used to assess every construct considered. The most important measurement properties of PROMs according to COSMIN approach are: validity, responsiveness, and reliability. To evaluate these psychometric properties, the study was divided into four steps: identification, prioritization, assessment, and summary of all PROMs considered in patients with mPCa.

Following identification of the instruments and the corresponding trials, which are listed in Appendix A, a systematic review of the literature was performed to identify any relevant PROM development or validation studies evaluating the psychometric properties of each instrument. Only studies up to September 2021 were considered and assessed. Only PROMs that had been applied or utilized in >1 study were evaluated. Databases searched included MEDLINE (PubMed). The PROM filter developed for PubMed by the Patient-Reported Outcomes Measurement Group, University of Oxford, and a highly sensitive and valid search filter template, developed by COSMIN to identify studies concerning measurement properties, were used. An example of this comprehensive search strategy is included as Appendix A.

### 2.2. Inclusion and Exclusion Criteria

The following inclusion criteria were utilized: studies that reported the development and evaluation of the measurement properties of a PROM (development and validation papers); studies with a population of >80% of patients with mPCa; and full-text articles published in the English language only. Studies including participants with other disease types were considered as long as the results for patients with mPCa were reported separately. Studies that concerned the application of a PROM measure or that used the instrument as an outcome measure were excluded, as well as studies assessing non-health related PROMs (e.g., treatment satisfaction) and meeting abstracts, conference abstracts, editorials, and commentaries.

### 2.3. Screening

In both systematic reviews, abstracts and full texts were screened independently by two members of our research team (A.D., E.S.S.), who undertook all stages of the process following the original electronic search, working independently of each other, and then comparing outcomes at each stage of the process. Any disagreements were resolved by discussion between all the members of the research team. To identify the final articles for inclusion in the review, full-text papers were further explored independently by the reviewers. The results from the database search and the study selection process are presented in Appendix A.

### 2.4. Data Extraction

Data from each study of each PROM were extracted for the following characteristics: information on the instrument acronym, the core domain(s) measured, area assessed and number of data items, a brief description of the PROM, the recall period, information relating to scoring, estimated completion time and mode of administration, licensing information, the number of studies evaluating the instrument and the highest COSMIN rating. This information is reported in summary cards for each PROM (Appendix A).

### 2.5. Appraising Methodological Quality

The methodological quality of each of the included studies was assessed according to the COSMIN checklist [9]. To evaluate the methodological quality of studies, the COSMIN 4-point checklist was used. This checklist shows nine measurement properties categorized into three overarching domains concerning validity (content validity, structural validity, hypothesis testing, cross-cultural validity, and criterion validity) reliability (internal consistency, reliability, and measurement error), and responsiveness. Table 1 shows the detailed definitions of each measurement property included.

Only boxes with the measurement properties evaluated in the study were completed. Eligible studies were also rated according to a four-point rating scale indicating “very good”, “adequate”, “doubtful”, or “inadequate” methodological quality. In addition, an overall rating of methodological quality for each assessed measurement property was calculated per box according to the “worst score counts” method [4]. Only thereafter were all results per study measurement property of a PROM quantitatively pooled or summarized against the criteria for good measurement properties to obtain the overall ratings.

### 2.6. Reporting of Psychometric Results

All PROMs considered were graded on the quality of the evidence as previously reported [6]. The quality of evidence for structural validity was the starting point for determining the quality of evidence for internal consistency. However, Cronbach’s alpha was difficult to interpret since it is not based on a unidimensional scale [10]. Internal consistency is based on the available evidence for structural validity, while the prerequisite for the interpretation of internal consistency is the unidimensionality. The COSMIN manual for systematic PROM reviews recommends ignoring the results of studies on internal coherence of scales that are not one-dimensional. Content validity is the most important psychometric property for the COSMIN methodology because it reflects if the items of the PROMs are “relevant”, “comprehensive”, and “comprehensible” to the construct of interest and study population [4]. Therefore, we considered this as an important property and used it as a threshold for inclusion, e.g., if a PROM did not demonstrate adequate validity of the content for the domain of interest in our target population (mPCa), then it was excluded from the study [5]. The reviewers assessed, through a subjective judgment, the adequacy of the initial qualitative work in the target population to identify the constructs of importance and then to assess the validity of the content. This process included the PROM development study, the quality and results of further validity studies of the content on the PROM (if available), and a subjective evaluation of the content of the same [9].

### 2.7. Appraisal of Levels of Evidence

To determine the overall quality of each measurement property, an assessment of the levels of evidence appraisal was carried out. This process produced a final evaluation for each PROM for each measurement property as previously reported [11]. The PROMs were grouped into three categories to allow for an evidence-based recommendation: (A) PROMs with evidence for sufficient content validity and at least low-quality evidence for sufficient internal consistency. PROMs rated with (A) can be recommended for use and results can be trusted; (B) PROMs that have potential to be recommended but they require further research; (C) PROMs with high-quality evidence for an insufficient measurement property. Finally, to determine whether, overall, the “relevance”, “comprehensiveness”, “comprehensibility”, and content validity of the PROMs are “sufficient”, “insufficient”, or “indeterminate”, all the identified studies for each PROM were qualitatively summarized. The overall ratings were accompanied by a grading for the quality of the evidence and “high” (+), or “moderate” (±), or “low” (?), or “very low” (−) were the levels of evidence rating. Table 2 presents the levels of evidence criteria.

## 3. Results

Overall, our systematic review identified only seven PROMs utilized in RCTs that focused on patients with mPCa. The most frequently used PROMs in RCTs of patients with mPCa were the Functional Assessment for Cancer Therapy—Prostate (FACT-P; *n* = 18), the Brief Pain Inventory—Short Form (BPI-SF; *n* = 8), and the European Organization for Research and Treatment of Cancer quality of life core 30 (EORTC QLQ-C30; *n* = 6). Each PROM identified in the systematic review and the corresponding publications are listed in Appendix A.

A subsequent systematic review was undertaken to identify publications developing the tools or evaluating the psychometric properties of PROMs. The second search related to the PROMs returned a total of 283 records, identified using PubMed, divided as follows: EORTC-QLQ-C30 (*n* = 67), FACT-P (*n* = 65), BPI (*n* = 62), EORTC-QLQ-PR25 (*n* = 39), BPI-SF (*n* = 23), EQ-5D-5L (*n* = 2), and BFI (*n* = 15). After the removal of duplicates and abstract screening, 12 articles were included and examined through the full-text assessments. Among those, only five full texts were selected for further examination, met the inclusion criteria, and were evaluated according to the COSMIN checklist (Table 3; Figure 1). The characteristics of the studies included are presented in Appendix A. At the end of the screening phase we identified only one generic questionnaire about cancer quality of life: the Functional Assessment for Cancer Therapy—Prostate (FACT-P) [11] and two specific questionnaires concerning a specific subdimension of the concept of quality of life, the pain: The Brief Pain Inventory (BPI) [12] and the Brief Pain Inventory—Short Form (BPI-SF) [13]. The FACT-P is a questionnaire usually used in clinical practice with cancer patients 18 years and older to assess quality of life. The FACT-P and the BPI-SF were the most evaluated questionnaires in validation studies (*n* = 2, respectively), followed by the BPI (*n* = 1). A summary of the number of items and concepts assessed for each of the PROMs evaluated is listed in Appendix A.

Table 4 presents the COSMIN checklist scores evaluating the methodological quality of each study and reporting the assessment of the measurement properties per PROM. Structural validity was evaluated in all studies, but only a few of them reported the results concerning this property. Reliability and hypothesis testing were evaluated in four studies closely followed by internal consistency (three studies). Criterion validity and responsiveness were less frequently reported (two studies), and no study evaluated measurement error and cross-cultural validity as psychometric properties. Content validity was assessed for each study, in line with the published COSMIN guidelines. This rating was provided by the reviewers and was thereby partly subjective in nature. Content validity, internal consistency, and responsiveness were the best performing properties, with three of the five studies receiving a score of “very high”. Regarding the content validity, two of the five studies received a score of “high”, while for responsiveness, two out of the five studies received a score of “very high”. The internal consistency was presented as Cronbach’s α, while the reliability was calculated using the Interclass Correlation (ICC). Regarding reliability, two studies gained a score of “very high”, and the other two gained, respectively, “high” and “moderate”.

Hypothesis testing also scored well with four of the five studies attaining a “very high” grading in two studies and a score of “high” and “very low” in the last two.

Responsiveness was assessed in two of the studies and achieved a grading of “very high” and “high”, while criterion validity was well assessed, achieving two of the five studies with a score of “moderate”. The property with the worst performance was the structural validity, where three studies obtained a score of “very low”, and two studies received a score of “moderate”. Structural validity was assessed by exploratory factor analysis only in two studies.

The FACT-P and the BPI showed a high content validity, while the BPI-SF showed a moderate content validity. BPI and FACT-P instruments showed a high internal consistency (summarized by Cronbach’s α above 0.70 but not higher than 0.95) and they were categorized as A. On the other hand, BPI-SF was categorized as C because of its lack of psychometric evidence.

## 4. Discussion

Our study aimed to assess the psychometric properties of the identified PROMs to provide recommendations to clinicians and researchers regarding which is the most appropriate instrument for assessing health-related quality of life in patients with mPCa. The psychometric properties of each PROM adopted in RCTs focusing on patients with mPCa were rigorously assessed using the COSMIN methodology. Overall, three instruments measuring quality of life and health status were identified as being developed or validated in one or more studies focusing on patients with mPCa. Overall, only five studies reporting the psychometric properties of PROMs met the inclusion criteria. However, none of the investigations met all COSMIN standards for methodological quality. When implementing the “best fit” strategy and considering the lack of available evidence, the results of our systematic review suggest that the BPI, BPI-SF, and FACT-P should be recommended for the assessment of patients diagnosed with mPCa. Specifically, the BPI is a self-report instrument designed to evaluate the severity of pain and the impact of pain on daily functions in cancer and other diseases [12]. The BPI also asks the patient to indicate the percentage of relief provided by pain treatments or medications. The adoption of the BPI-SF, which is used for clinical monitoring of pain and in clinical trials where multiple assessments are required, should be considered for its brevity and for the patient’s ease of use [12,13]. The FACT-P is a disease-specific questionnaire used to assess QoL in patients with PCa and asks about symptoms and problems specific to the disease itself [11]. This disease-specific questionnaire should be considered for the assessment of QoL in mPCa.

Although, our systematic review identified seven PROMs used in RCTs on mPCa, studies assessing the psychometric properties were not available for three of them. For example, the EORTC QLQ-C30, the EORTC QLQ-PR25, and the EPIC are recommended by ICHOM and are widely adopted specific questionnaires used to assess the quality of life of mPCa patients [17]. These PROMs were recently recommended for the evaluation of men with clinically localized disease due to their rigorous methodological development and characterization processes in the setting of localized PCa. Similarly, other consortia such as the ICHOM propose different tools (namely, the EORTC and EPIC). However, from a methodological standpoint, there were no data available regarding their psychometric properties to confirm the quality and validity in the mPCa setting. Such a lack of data highlights the necessity for the development of further studies assessing the psychometric properties of commonly adopted PROMs to inform recommendations on instrument use not only for a specific disease but also according to the staging (localized vs. metastatic PCa).

Our results should also be considered in the context of the limited evidence published in the literature. Indeed, only three out of the seven PROMs identified had limited data available on their psychometric properties. For example, measurement error and the cross-cultural validity were poorly assessed with none of the included studies reporting these measurement properties. The current lack of measurement error reporting across all PROM instruments affects our interpretation of the consistency of each measure, as we cannot be confident that any observed change is not attributable to background “noise” [18]. On the other hand, the lack of cross-cultural validity may threaten the external validity of the PROMs considered in settings that are different than the development cohort.

On the other hand, the other psychometric properties have all been well evaluated. First, internal consistency and criterion validity were assessed in two PROMs out of three. The ratings obtained provide evidence to support the validity of the defined subscales of each instrument and provide evidence to support the consistency of measurements with gold comparison standards. Second, reliability was one of the best measurement properties, which was well evaluated in all recommended PROMs. These results are promising, as accurate and reproducible measurements are a prerequisite for classifying high-quality PROM instruments [19]. Similarly, structural validity and content validity were well evaluated in each of the PROMs identified. The latter is the most important measurement property according to COSMIN guidelines and concerns the relevance, comprehensiveness, and comprehensibility of each instrument [10]. Hypothesis testing was a good performing measurement property, too. Finally, responsiveness was well assessed in two PROMs out of three, and therefore we can make assumptions about this tool and its ability to detect change over time [20].

Based on the results of the analysis of the individual psychometric properties of each PROM and adhering strictly to the COSMIN methodology, the BPI-SF presents evidence assessment only for five out of nine of the psychometric properties considered, and half of these show an insufficient score after the evaluation based on COSMIN methodology. Thus, the use of this questionnaire is not recommended because of its lack of sufficient psychometric properties, despite the authors of the questionnaire recommending its use for its brevity and ease of use for the patients. The BPI and FACT-P showed evidence of sufficient content validity (any level) and very high-quality evidence of internal consistency and should be recommended for use in clinical practice and in the research setting. BPI is a questionnaire used to assess pain, which is one of the dimensions that defines the quality of life in patients with cancer. However, quality of life is not based solely on the perception of pain, and it therefore seems useful to recommend the use of this questionnaire together with a more specific questionnaire for this aspect. The FACT-P is a specific questionnaire on PCa, used to assess the quality of life in general, and its use is appropriate for the clinical evaluation of metastatic patients.

Some limitations of our study should be highlighted. Firstly, the search was restricted to the English language only, which may have introduced a language bias. Our aim was to compare the psychometric aspects of health-related quality of life instruments, and there is no evidence to suggest that this objective was influenced by the language restriction. Second, although the reference lists of included articles were comprehensively checked to identify any further relevant studies, there is the potential that some may have been missed. Another limitation concerns the exclusive use of PubMed in research. This may have influenced the number of articles found and, therefore, also the subsequent assessment. Moreover, the number of studies identified for each PROM was small, and therefore the results of the analyses carried out relate exclusively to a few articles. This could undermine the generalization of our results. On the other hand, the number of questionnaires was selected on the basis of specific criteria and this also entails greater accuracy in the selection and validity for scientific purposes. Finally, although the systematic analysis of the included articles was carried out strictly adhering to the COSMIN guidelines, the review process is partly subjective, so, for this reason, the authors ensured the involvement of more than one reviewer to ensure consensus and consistency.

## 5. Conclusions

There are several generic and disease-specific PROMs that can measure HRQoL of patients with mPCa both in the research and clinical setting. Although the number of studies assessing the measurement properties of each instrument is limited, there is sufficient evidence to support the use of the BPI and FACT-P in patients with mPCa. We recommend the use of the FACT-P, which provides a broader assessment of QoL and wellbeing for the clinical evaluation of patients with mPCa.

## Figures and Tables

**Figure 1 cancers-14-05120-f001:**
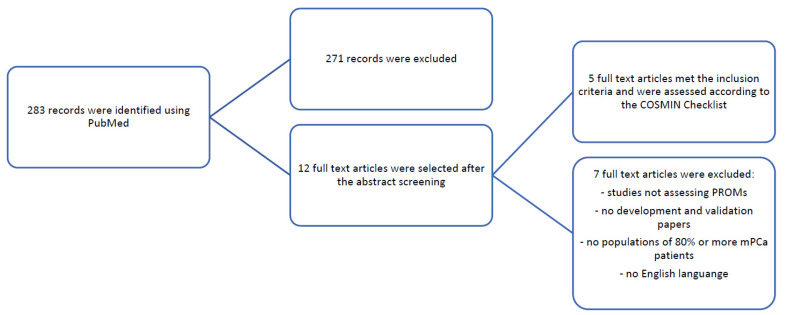
Flow diagram for study selection.

**Table 1 cancers-14-05120-t001:** COSMIN definitions of domains, measurement properties, and aspects of measurement properties.

Measurement Property	Definition
Reliability	The proportion of the total variance in the measurements that is due to “true” † differences between patients. Usually expressed as an intraclass correlation coefficient (ICC).
Internal Consistency	The degree of the interrelatedness among the items. Usually expressed as Cronbach’s α, which measures the extent to which items in a PROM (sub)scale are correlated.
Measurement Error	The systematic and random error of a patient’s score that is not attributed to true changes in the construct to be measured. The difference between the true or actual value and the measured value.
Content Validity	The degree to which the content of a PROM is an adequate reflection of the construct to be measured. Reflects whether all items of a PROM are relevant, comprehensive, and comprehensible for the population of interest and requires an element of subjectivity when assessing.
Structural Validity	The degree to which the scores of a PROM are an adequate reflection of the dimensionality of the construct to be measured. Examines the extent to which the underlying putative structure of a scale is recoverable in a set of test scores.
Cross-cultural Validity	The degree to which the performance of the items on a translated or culturally adapted PROM are an adequate reflection of the performance of the items of the original version of the PROM.
Hypotheses Testing	Item construct validity. Concerns the degree to which the scores of an instrument are consistent with hypotheses based on the assumption that the instrument validly measures the construct to be measured.
Criterion Validity	The degree to which the scores of a PROM are an adequate reflection of a “gold standard”. Criterion validity is an estimate of the extent to which a measure agrees with a gold standard.
Responsiveness	The ability of a PROM to detect change over time in the construct to be measured. Refers to the ability of an instrument to distinguish clinically important changes as the result of an intervention.

† The word “true” must be seen in the context of the CTT, which states that any observation is composed of two components—a true score and error associated with the observation. “True” is the average score that would be obtained if the scale were given an infinite number of times. It refers only to the consistency of the score and not to its accuracy. Interpretability is not considered a measurement property but an important characteristic of a measurement instrument.

**Table 2 cancers-14-05120-t002:** Definitions of quality levels.

Quality Level	Definition
High	We are very confident that the true measurement property lies close to that of the estimate * of the measurement property
Moderate	We are moderately confident in the measurement property estimate: the true measurement property is likely to be close to the estimate of the measurement property, but there is a possibility that it is substantially different
Low	Our confidence in the measurement property estimate is limited: the true measurement property may be substantially different from the estimate of the measurement property
Very low	We have very little confidence in the measurement property estimate: the true measurement property is likely to be substantially different from the estimate of the measurement property

* Estimate of the measurement property refers to the pooled or summarized result of the measurement property of a PROM. These definitions were adapted from the GRADE approach.

**Table 3 cancers-14-05120-t003:** Overview of the PROM instruments extracted after the screening procedure.

Prom	Full Name	Abstract Screening	Full-Text Screening	Full-Text Extracted
FACT-P	Functional Assessment of Cancer Therapy—Prostate Cancer	**65**	**3**	**2**
EORTC QLQ-C30	EORTC QLQ Quality of Life Questionnaire	**67**	4	0
BPI	Brief Pain Inventory	**62**	3	1
BPI-SF (Short Form)	Brief Pain Inventory Short Form	**23**	2	2
EORTC QLQ-PR25	EORTC Prostate Cancer Module	**39**	0	0
EQ-5D-5L	EuroQol five dimensions, five levels questionnaire	**12**	0	0
BFI	Brief Fatigue Inventory	**15**	0	0

**Table 4 cancers-14-05120-t004:** COSMIN Checklist scores evaluating methodological quality of each study per measurement property and PROM.

	Authors	Content Validity *	Structural Validity *	Internal Consistency *	Reliability *	Measurement Error *	Hypothesis Testing *	Responsiveness *	Cross-Cultural Validity *	Criterion Validity *
FACT-P	Clark et al., 2014 [14]	Very high	Very low	Very high	High	/	Very high	Very high	/	/
	Robinson et al., 2013 [15]	High	Moderate	Very high	Very high	/	High	/	/	Moderate
BPI-SF	Clark et al., 2014 [14]	Very high	Very low	/	Moderate	/	Very low	Very high	/	/
	Gater et al., 2011 [16]	Very high	Very low	/	/	/	/	/	/	/
BPI	Robinson et al., 2013 [15]	High	Moderate	Very high	Very high	/	Very high	/	/	Moderate

* See Table.

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
