# Peer review of "A Systematic Review to Evaluate Patient-Reported Outcome Measures (PROMs) for Metastatic Prostate Cancer According to the COnsensus-Based Standard for the Selection of Health Measurement INstruments (COSMIN) Methodology"

_cancers, 2022, doi:10.3390/cancers14205120_

Round 1
Reviewer 1 Report (Previous Reviewer 4)
The manuscript has been improved and may be published after minor polishing:
References are not in the same format (e.g. DOI is provided only for some articles)
Upper and lower cases should be unified in Keywords.
Appendix A has become Supplemental Material, however, it is still in the main manuscript.
Author Response
Please see the attachment.

Reviewer 2 Report (Previous Reviewer 2)
Authors addressed our concerns
Author Response
Please see the attachment.

This manuscript is a resubmission of an earlier submission. The following is a list of the peer review reports and author responses from that submission.
Round 1
Reviewer 1 Report
Authors should be congratulated. The paper is well written, clear and concise. Methodology is unremarkable. The main message of the study is that PC is yet not enough studied. The limited availability of standardized, reproducible, and reliable questionnaires to assess patient reported outcomes for mPCA is a clear demonstration. Therefore, what has been considered a weakness of the study is indeed a strength that should be expanded in the discussion/conclusion section (up to now limited to only a brief sentence, line 277)
Reviewer 2 Report
Written very well and clear
the major drawback is the small number of studies included
Reviewer 3 Report
Thank you for the opportunity to review this article.
In this review article, the authors with affiliation at the PIONEER Consortium assess the literature to determine the use of PROMs in RCTs, as recommended by the COSMIN criteria.
Using the strict criterias defined in the study, only 5 papers were ultimately included in the analysis. The study highlights the inconsistencies and the deficiencies of current PROMs in RCTs involving patients with metastatic prostate cancer.
It is a nice overview of the current status of PROMS reporting, and I do not believe any any changes are required.
Thank you kindly.
Reviewer 4 Report
Reviewer agrees with the importance of patient-reported outcomes measures (PROMs) as explained in the Introduction: "A better understanding of how both the disease and the treatments affect the quality of life (QoL) is essential to guide the improvement in patient care. In addition to oncological endpoints and clinicians’ reported outcomes such as progression or development of a castration-resistant state, Patient-reported Outcome Measurements (PROMs) are important to assess the patient-subjective experience with disease and treatment. Moreover, they can provide clinically relevant, patient-focused, and reliable perspectives on quality of life and patient symptom experience." However, striking issues have been found:
Both abstract and manuscript are not in the standard format. For example, sections Results and Discussion are missing, while Conclusions are on two pages.
A total of 283 abstracts were screened and 12 full-text studies were evaluated. References even for the 12 full-text studies are provided only as DOI in supplementary table 1. Among those, five full-texts were selected for further examination and met the inclusion criteria and were evaluated according to the COSMIN Checklist (Fig.1). These five studies (in fact only three - Clark et al 2014, Robinson et al. 2013 and Gater et al. 2011; see Table 2) are missing in the References!?
Reference is also missing for the "Functional Assessment for Cancer Therapy (FACT). Summary Card of the FACT-P (for prostate cancer) is provided only as Supplementary Table 2 which is currently mentioned in Methods. This should also appear in the Results and a typical questionnaire for patients should also be provided (e.g. https://www.facit.org/measures/FACT-P).
Information provided in Appendices should be converted either to Tables or Supplements.
In Table 2, there should be a reference to the information now provided in Appendix B.
A few typo errors were found:
Abstract - Abbreviation RCTs not explained at all. Other abbreviations are only defined in the main text (e.g. FACT-P, BPI). mPCa is sometimes as mPca.
Table 2 - Robinson 2013 or 2003 ?
Page 9 - different fonts
Appendix B - different fonts